# A Grounded Theory Approach to the Influence Mechanism of Residential Behavior among Mongolian Yurt Dwellers in China

Jingwen Che [1], Wanqing Su [2], Liyan Bai [2,3,*] and Hong Guo [1]

1. School of Civil Engineering and Architecture, Shaanxi University of Technology, Hanzhong 723000, China; cjw2021@snut.edu.cn (J.C.)
2. School of Architecture, Harbin Institute of Technology, Key Laboratory of Cold Region Urban and Rural Human Settlement Environment Science and Technology, Ministry of Industry and Information Technology, Laboratory of National Territory Spatial Planning and Ecological Restoration in Cold Regions, Harbin 150006, China
3. College of Architecture, Inner Mongolia University of Technology, Hohhot 010000, China
* Correspondence: naomugeng@163.com

**Abstract:** The residential Mongolian yurt is representative of nomadic culture and its mobile residences. Behavior forms the closest connection the environment and people. There is almost no research about the residence and lifestyle of Mongolian yurt dwellers from the perspective of housing behavior. This study applied grounded theoretical methods to analyze the theoretical model of the influence mechanism of Mongolian yurt dwelling on behavior. Interviews were conducted according to the principle of purposive sampling, and were summarized in five categories: production lifestyle, natural environment, residential characteristics, cultural beliefs, and emotional experience. Production lifestyle is the core category and the critical factor in dwelling behavior, which interacts with the other main categories. Pastoral policy is the factor that has worked most quickly and directly to influence housing in yurts. Mongolians living in yurts on the prairie show higher satisfaction compared to those in urban housing, which is related to the Mongolian advocacy for nature and freedom.

**Keywords:** residential behavior; Mongolian yurts; influencing factors; grounded theory

## 1. Introduction

Mongolian yurt dwellers are typical representatives of nomadic culture and mobile residential life. Shuji Funo pointed out in *World Residence*: "The yurt is one of the best mobile dwellings created by humans" [1]. Rapopor researched the world's dwellings and proposed that the most sophisticated tent is the yurt, and the yurt is a symbol of mobility; Mongols fully control their way of living and living freely [2]. With the changes in the social background of the new era, under the guidance of Inner Mongolia's pastoral policy, the way of life has shifted from "living by water and grass" to settlement. This has brought about the current status of herdsmen's housing and housing needs, causing the unique nomadic civilized life to face a crisis of elimination. However, the current research efforts on yurts are mostly focused on space [3,4], physical environment [5,6], structure and materials [7], cultural implications, inheritance and development [8], and optimization design [9–11].

The earliest research on the yurt is mainly the notes of travelers from European and American countries [12]. It comprises a comprehensive interpretation of the traditional yurt, including the origin and development [13], structural system, production techniques, aesthetic characteristics, and life customs [14,15]. Studies have focused on the cultural implication of the yurt, including the order of the inner space, the order of inferiority, the order of sacred customs, and the concept of time and space [16,17], as well religious beliefs, natural views, values, social concepts, and life customs [18–20]. Japanese scholars such as Hairihan, Rie Nomura, and Koki Kitano have researched contemporary Mongolian life and changes. Their research reveals that nomads have a high degree of understanding of the

natural environment and protection and inheritance, besides Mongols having a high degree of satisfaction with the nomadic lifestyle and a high degree of community awareness [21]. After settlement, their production and lifestyle, residence, customs and relationships have changed, showing the law and characteristics of evolution [22–25], and there have been two trends in the simplification and continued use of the traditional yurt [26]. In terms of the spatial layout, structure and use of settled residences, the space composition, space order, space concept, and methods of using the yurt in different periods have been continued [27].

Residence behavior is the concrete manifestation of residential life, and analyzing it is the most direct and effective way to study the space utilization and housing needs of yurt dwellers. The current residential behavior research projects include those on residential behavior and space, the environment, influencing factors, and applications. The correspondence between residential behavior and space has the characteristics of comfort, functionality, expressiveness, variability, and health [28]. The most basic task of space is to provide a place for various daily activities and to meet the specific needs of various activities. Therefore, space shaping and physical behavior are interrelated and restrict each other [29]. Various characteristics of space, including proportion, scale, color, permeability, and overall aesthetic effect, have an impact on human behavior [30]. In addition, pandemic diseases also have an impact on the changes in residents' behavior and housing needs. The COVID-19 epidemic puts forward new requirements for housing health and adaptability [31,32]. The influencing factors of residential behavior are the best entry point to study its generation, law, and evolution, as well as the key medium for studying the elements of people, space, and built environments. They involve diversified levels, including different types of people, residences, and regions. A housing study of 11 Alzheimer's groups in Finland showed that the factors affecting life and behavior include physical, space, business, and personal factors [33]. Various attributes of residents, such as family structure, residence time, economic status, etc., will have a positive or negative impact on life behavior [34]. Another study revealed 8 factors that affect the family's living lifestyle: basic living awareness, hobbies, usage details, attention to the environment, etc. [35]. Environmental factors and daily habits greatly influence the behavior of residents [36]. Based on the process of change and group characteristics, the factors affecting residential behavior are summarized as physiological factors, psychological factors, natural factors, and social factors [37,38]. A large number of studies are on the issues of residential behavior and energy consumption [39–41].

In behavioral research, many scholars also reveal the links related to behavior from sociology, while others are in business and medicine. Current researchers focus on the development and application of behavioral models [36,42,43], which include the PMV comfort model from the perspective of building energy consumption and behavior interaction [43], BDI [44], ABMs [45], etc. Some new methods, such as UWB [46], VR technology [47], and acquisition of geospatial information based on ArcGIS and Gephi, are used in the architecture field [48]. However, compared with real space, it is difficult to obtain a sense of freshness, width, and depth in a virtual space [49,50]. Furthermore, virtual behavior and spatial perception are different from real sensory experience effect feedback, and technical methods often ignore the influence of human subjectivity and emotional willingness [51]. Grounded theory is a sociological qualitative research method that focuses on the social phenomena of people and the environment. It can be local and substantive, as well as offering extended and formal theoretical results [52]. At present, it is widely used in economics, psychology, education, medicine, etc. Applications in the field of architecture involve acoustic environments [53,54], and nursing homes [55], urban spaces [56], underground spaces [57], educational spaces [58], and gardens [59].

Until now, there has been no special study on yurt dwelling behavior, especially the study of the relationship between the yurt and dwelling behavior, to explore yurt dwelling and its lifestyle. Therefore, this study applied grounded theory to construct a theoretical model of the factors affecting the residence behavior of Mongolian herders, analyzed the relationship between the residence behavior, environment, and Mongolian herdsmen, and

explored the development of yurts and residential life from the perspective of behavioral research. The structure of this article is as follows: Section 2 sorts out the data of the participants collected in the interview. In Section 3, five categories are summarized and the relationship between them is explained. Section 4 discusses the relationship between the various categories. The paper ends with some conclusions and ideas for further works.

## 2. Materials and Methods

### 2.1. Participants and In-Depth Interviews

At present, only a few areas in Inner Mongolia, such as Bayanwindur Sumu, Zarut Flag, and Ulagai in Tongliao City, still retain nomadic life. The yurt residence in the summer camp of Zalut Banner was selected for observation and participatory life surveys. The "original data" of surveying and mapping and interview data were obtained to recognize the current situation and changes of Mongolian herders' living in a modern context. During the survey process, we interviewed 33 respondents between 20 and 71 years old, 7 women and 26 men. According to their familiarity with yurts, 27 interviewees with yurt living experience were selected, and 6 interviewees had a certain degree of understanding and research on yurts. Those interviewees had different identity backgrounds, including 20 Mongolian herders, 3 Mongolian scholars, 3 yurt-related research architects, and 7 Mongolian students (one is a student from Mongolia), as shown in Figure 1.

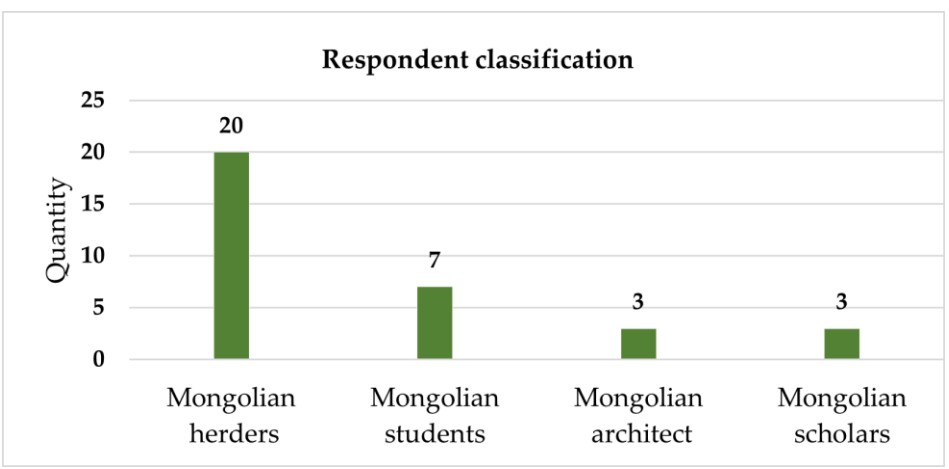

**Figure 1.** This is the respondent classification.

According to Lincoln and Guba, the number of samples for grounded theory should exceed 12. A total of 34 interviewees were selected for formal interviews in this study, meeting the requirements of "purposive sampling" in qualitative research [60]. The above sample populations are representative and meet the theoretical sampling standards. In this study, a semi-structured interview format was selected to conduct in-depth interviews with the interviewees, obtain more sufficient basic data on the subject, and organize the interviews with audio recordings. The interview time for each respondent was 20–90 min, with an average of 60 min. The content of the interview was based on the theme of "life behavior, living conditions, awareness, expected residence, an interviewee's background", and the survey outline containing the scope of the survey content and several major questions was established in advance. The interviewer could adjust flexibly according to the actual situation of the interview.

According to the standard procedure of the grounded theory (Ellis, Strauss, and Corbin 1992) [61], the interview data was converted into an electronic text, and the data were coded and analyzed in the next step. This study used Mindjet MindManager software, which can effectively improve work efficiency and classify non-numerical and non-structured data, to analyze and classify them [54].

### 2.2. Data Coding Analysis

The grounded theory uses a variety of data collection methods under natural circumstances and uses researchers as research tools to conduct a holistic exploration of social phenomena. It mainly uses inductive methods to analyze data and form theories. It gains explanatory understanding by interacting with the research object to construct its behavior and meaning [62]. This method includes two parts: data collection and data coding, which provides clear method guidance for researchers [63]. The research methodology flowchart is shown in Figures 2 and 3. Specific steps are shown in Table 1, and are as follows:

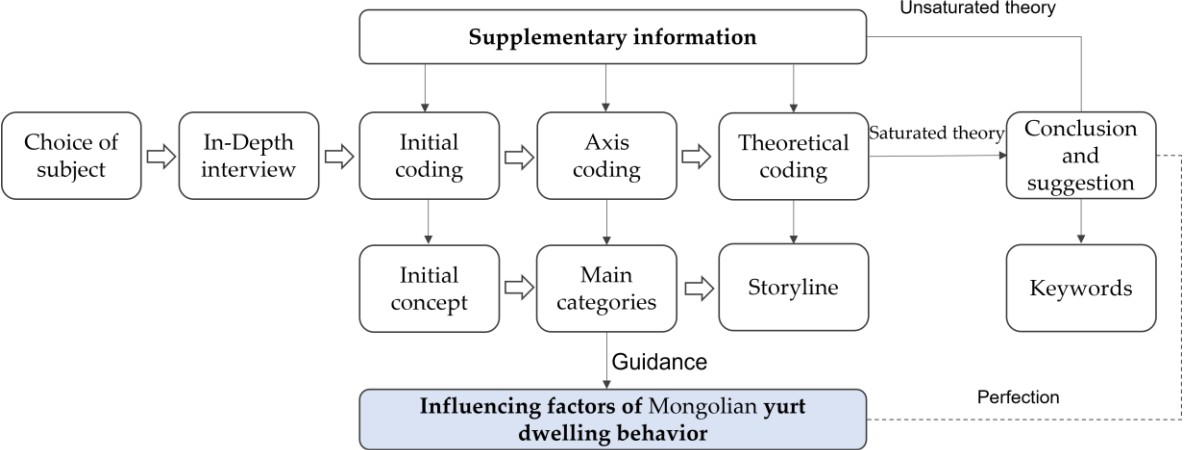

**Figure 2.** Step chart of grounded theory.

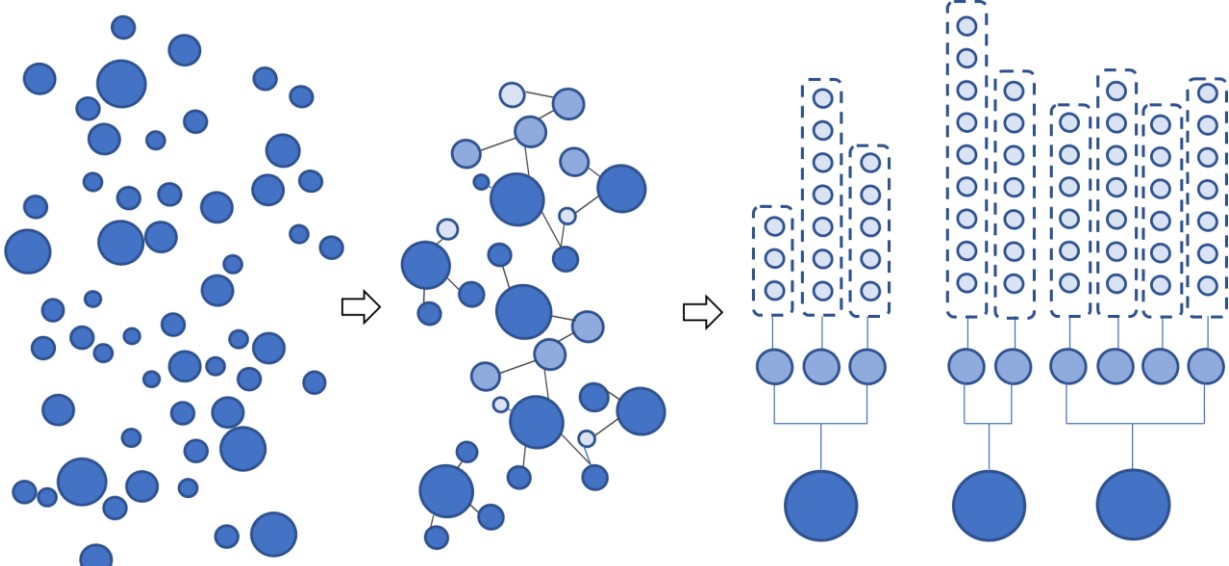

**Figure 3.** Schematic diagram of a flowchart.

(1)    Screening of interviewees: A purposeful sampling of participants—helpful insights into research questions, including from occupants and researchers, and people with a certain degree of familiarity;

(2)    In-depth interviews: Ask questions based on research questions and interview outlines. The purpose is to stimulate thinking and dig deeper into the interview information;

(3)    Open coding: Based on the yurt residence behavior influence mechanism as the center, find out the concept and characteristics of the data;

(4)  Axis coding: Combine similar viewpoints to form categories. For example, the production–lifestyle categories are similar concepts based on the initial coding, and other categories are composed of concepts from other classifications;

(5)  Acquisition of subcategories: To obtain a simple and clear conclusion, further comparison, induction, and integration of the category "Ax, Bx . . . " obtained by the coding, extracted 22 subcategories, represented by numbers, as shown in the following Table 1;

(6)  Theoretical coding: Summarize the main categories through comparative analysis; these main categories cover most categorized concepts;

(7)  Linking categories: Establishing links between the main categories and analyzing how the main categories affect each other;

(8)  Discover core categories, such as production–lifestyles in core categories that have extensive relationships with other categories;

(9)  The substantive theory is produced: the theory about the factors influencing the yurt dwelling behavior.

**Table 1.** Coding process for open coding, axial coding, and selective coding based on GT (aaX: labeling data code number; aX: conceptualizing data; AX: categorizing data code number; AAX: categories code number AA1-1).

| Sorting Memos | Labeling | Conceptualizing Data | Categorizing Data | Subcategories | Categories |
|---|---|---|---|---|---|
| ("How is your living condition in the yurt?") The yurt I live in is a traditional yurt, which is the kind of wooden frame. During that time, my dad and I lived in it. When I woke up every morning, it was very humid, and the quilts were all wet. When it comes to summer, it is to lift up the felt wrapped around Hana at the bottom, and to open it is to lie down and watch the sheep. Make a fire by yourself, and usually listen to the radio. . . . Original text of interviews with 33 people | aa1 The yurt I live in is a traditional wooden yurt. aa2 Live with my dad. aa3 I wake up very humid every morning. aa4 In the summer, it is to lift up the felt that wraps Hana with the bottom edge. aa5 lie down and watch the sheep through that. aa6 Make a fire by yourself . . . . . . A total of 822 labels | a1 Live in a traditional yurt. (aa1) a2 The traditional yurt is made of wood. (aa1) a3 Traditional yurts are humid in the morning (aa2) a4 Lift up the felt in the summer and look at the cows and sheep through the bottom (aa5, aa6) a5 I made a fire by myself, no electricity. . . . A total of 377 Conceptualizing data | A1 Residence experience (a1) A2 Traditional yurt material (aa2) A2 Traditional yurt humidity (aa2) A3 Productive activities (a4) A4 Daily activities. (a4) A5 Entertainment and leisure. (aa7) A7 Festival activities. (a10, a11) A8 Sacrificial activities. (a15) . . . A total of 59 categorizing data | AA1. Production—lifestyle AA1-1 Pastoral policy AA1-2 ways to produce AA1-3 living habit AA1-4 Individual Differences AA2. Cultural belief AA2-1 Sacrificial festival AA2-2 Etiquette AA2-3 Experience skills AA2-4 Cultural concept AA2-5 Spiritual belief AA2-6 Material carrier AA3. Emotional experience AA3-1 Inclination to live AA3-2 Perceptual evaluation AA3-3 Emotional preferences AA4. Residential characteristics AA4-1 Yurt space AA4-2 Structure and material AA4-3 Physical environment AA5. natural environment AA5-1 Climate environment AA5-2 Pasture AA5-3 livestock A total of 19 subcategories | AA1 Natural environment AA2 Residential characteristics AA3 Production lifestyle AA4 Cultural belief AA5 Emotional experience 5 main categories |

Following the above steps, the data were divided into individual thoughts, events, and behaviors. They were classified and refined by analyzing the similarities and qualified differences of concepts, and gradually formed a perfect and refined category [58]. Once the category was fairly complete, selective coding began. Open coding is the process of decomposing, checking, comparing, and conceptualizing data. In axis coding, the goal is to determine the connections and intersections between categories. The core of the theoretical coding process is to select a core category and the main categories related to it.

## 3. Results

Through the above procedure, therefore, this study produced 59 classifications. Five main categories have been produced, as shown in the figure below. Production–lifestyle is the core category, and the other main categories include cultural belief, emotional experience, residential characteristics, and natural environment. The results for five core categories and 19 key words are shown in Figure 4.

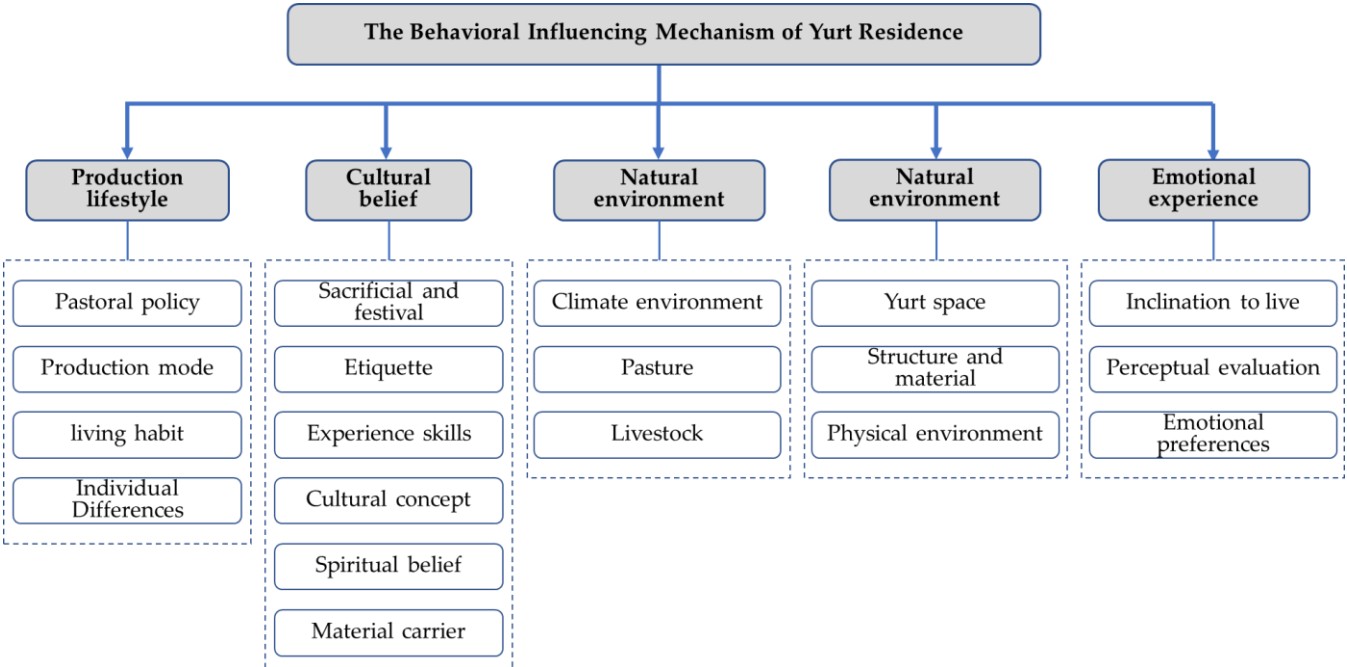

**Figure 4.** Structure of 5 core categories and 19 key words.

### 3.1. Production-Lifestyle

The characteristics of the category of production-lifestyle are composed of four subcategories: "pastoral policy", "production mode", "life habits", and "individual differences".

Pastoral area policy is the fastest-acting and most obvious reason for the change of production-lifestyle of pastoral dwellings. The herders mentioned that they could not graze freely, and only grazed in the designated pasture area due to the two pasture division policies in 1950 and 1997; according to the policy of agricultural cooperation from 1985 to 1964, downstream pastoral life turned to collective settlement life. Table 2 elaborates the changes in production methods. The living habit of residing in a yurt is compatible with the mode of production. It not only meets physiological needs and social needs, but its content, duration, and frequency are all based on the livestock and grazing conditions.

**Table 2.** Changes in production methods.

| Production and Lifestyle | Four Seasons Rotation | Rotary Animal Husbandry-Semi-Settlement | Rotational Animal Husbandry-Settlement | | |
| --- | --- | --- | --- | --- | --- |
| | | | Live in Community | Free Grazing | Pasture Division |
| Time | Before 1920s | 1940s (pasture division in 1950) | 1958–1964 agricultural production cooperative | 1967–1976 | 1980 grass and livestock double contract responsibility system | 1997 year pasture division |
| Residential form | ○ | ◎◇●△ | ●◎ | ◎ | ◎● | ●◎ |

○ Nomadic in four seasons, ◎ Bungalow in winter camp, ◇ Nomadic yurt in spring camp, ● Nomadic yurt in summer camp, △ Nomadic yurt in autumn camp.

The Mongolians "live by water and grass, and graze by water and grass", and carry out corresponding yurt dwelling behaviors around grazing. Buhe (a 25-year-old Mongolian architecture student) said: "In the summer, you can lift up the felt on the bottom of hana, and lie down to look after the cows and sheep". Zaragenbayr said: "The grassland has become smaller, the living conditions have changed, and the production methods are different so that the herders cannot cope with the nomadic life". As a result, the yurt will be ignored or abandoned for a period of time". Diagram and analysis of production–lifestyle categories is shown in Figure 5.

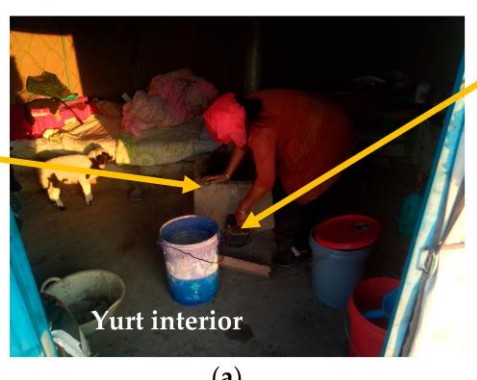

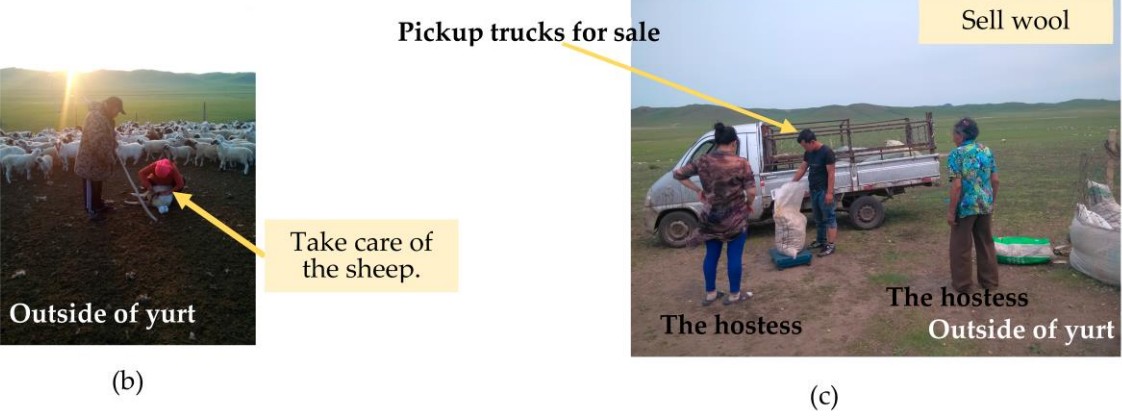

**Figure 5.** Diagram and analysis of production-lifestyle categories: (**a**) digging out dust; (**b**) taking care of cattle and sheep; (**c**) selling cattle.

The change of production-lifestyle is the most influential factor in the Mongolian living behavior. Alateng Aode (an architectural expert) mentioned: "Modern architects breaking the circular shape against the traditional production model, which is not good . . . The requirements of the production-lifestyle have led to the scattered settlement of

Mongolians". The 71-year-old horse racer herder said: "In the past nomadic period, they had to work and collect cow dung almost every day for the whole year. After finishing cleaning, wash your face and eat dinner. Then drive cattle and sheep, and come back from the mountain with firewood in winter". It can be seen that the production and lifestyle of the Mongolian people is the most direct and fundamental factor that determines the living behavior and living habits of the yurt dwellers.

Different behaviors and activities occur in the yurt due to individual differences, including gender, age, identity, tribe, and living experience. For example, according to the concept of elders and inferiority the seating positions of the elderly and guests are located in the north of the center; according to gender differences the female activity area is located in the east (left of door) direction of the center, and the male is located in the west (right of door) direction, as shown in Table 3.

**Table 3.** Space order normative behavior.

| Space Order Regulates Behavior | |
|---|---|
| The boundary between the holy and the vulgar | Male and female division |
| 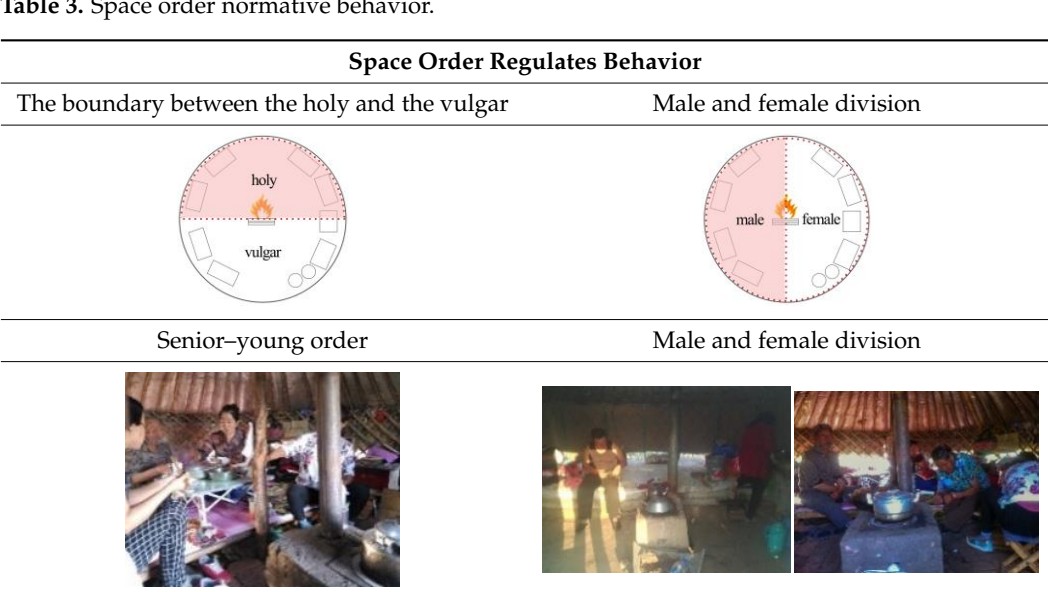 | |
| Senior–young order | Male and female division |

### 3.2. Cultural and Spiritual

Festival activities and sacrificial activities are an important part of Mongolian culture. Naadam is one of the most important festivals of the Mongolian people, and it is also a way of entertainment and socializing. Sacrificing Obo and fire worship are the most important sacrificial activities in Mongolian traditional festivals. Every year on the twenty-third day of the twelfth lunar month, fire sacrifices are held in the yurts. Breast meat with beef breastbone or sheep breastbone is cooked on a tunaga (a traditional, ornamented stove) or on a fire, a fire meal is served, and offerings and khatas are placed around the stove. This is followed by the chanting of fire sutras, chants, prostrations, and throwing of the breastbone and other offerings into the fire and the Heshig Hurtehu Ycsulal (a tribute distribution activity after a worship ceremony). In addition, there is usually a statue of Buddha in the northwest corner of the yurt, and the Buddha and ancestors will be worshiped on New Year's Eve. Fresh butter is added to the Buddha lamp and it is kept lit always. The next morning, the sheep's head cooked the day before is offered to the Buddha, and the whole family sits around and has a reunion dinner.

Moreover, there are sacrificial activities such as tree worship, mountain god worship, heaven worship, and so on. When talking about the process of offering sacrifices to the obo, the herder Uyoudai (a Mongolian herder interviewed) said: "Kill the sheep, and then put the dairy products. In the morning, go to the sacrifice, and then come back to hold the Naadam, horse racing, wrestling, and everyone will come to participate". Etiquette, customs, and rules in the yurt regulate people's words and deeds. Ayongga (a Mongolian college student) said: "You can't step on the threshold when entering the house. When you

sit in order from elders to children, young people like us can't sit in important positions. There are a lot of disciplines".

Mongolian spiritual beliefs originate from the space atmosphere, spatial form, nationality, and regionality of the Mongolian yurt, and penetrate into behavior, perception, and values. Ayongga said: "The circular plane shape is centripetal, which concentrates people's spirit. The dome of the yurt is also upward with tapered roof shape, which is a symbol of the combination of religion and life. Living in the yurt, [you] can feel the sense of connection in our faith". Mongolians worship primitive natural objects, admire the sky, respect nature, cherish natural resources, and focus on enjoying life now. Ayongga (Mongolian student) said: "I think the cultural heritage of each Mongolian is very important, among which spiritual belief is the most important . . . In the middle in Mongolian yurts is the fire, but it is not just heating. It makes sense to put it in the center of the yurt". Mongolians believe in Shamanism and Buddhism. Apart from worshiping the heavens and the Earth, they believe that fire symbolizes eternity and is the sustenance of life; they value horses and believe that horses have weak luck. In addition, Shaola (Mongolian herder) said: "The elders will also warn children not to play by the stove, not to step over the stove. It is not allowed to step on horse poles".

Thousands of years of nomadic life have allowed the Mongolians to accumulate a wealth of experience in grassland life and skills related to the composition, production, and construction of yurts. Alateng Aode said: "The construction of the yurt is determined by the long-standing nomadic lifestyle, which is inconvenient for modern society, but it is very convenient for the nomadic life". Material things such as the Mongolian yurt itself, Mongolian costumes, Mongolian food, and nomadic tools are the manifestations and inheritors of Mongolian culture. The yurt contains rich cultural connotations, including a shape like the sky, time-keeping, zodiac signs, and blessings and misfortunes. For example, in addition to bad weather conditions, you usually have to open the roof early to rise early, otherwise it means that unlucky things happen. People in yurts believe that "everything has animism" and believe in longevity, continuing the fetishism at the beginning of human civilization. A yurt is built according to the image of the imagined universe, and a yurt is a miniature of the world. Yurt dwellers believe in the sun god and face the door south or southeast, in order to avoid the northwest wind and feel more sunshine.

*3.3. Emotional Experience*

The characteristics of the emotional experience category are composed of three subcategories: "living tendency", "perceived evaluation", and "emotional preferences".

Emotional preferences and perceptual experience will affect the herdsmen's choice of housing and their feelings. Compared with the noisy, high-rise urban environment, Mongolian herders generally like to live in a free, relaxed, wide-view, and quiet pastoral area. Mongolians pay particular attention to sensory experiences. Alateng Aode said: "The circular space gives people a sense of solidity and belonging . . . I can sleep very deep in a circular space . . . The yurt gives a sense of intimacy and quietness. It is easy to perceive subtle changes in the surrounding environment in the yurt". For the Mongolians, the yurt is not just a three-dimensional form, but a complex including the smell, color, vision, and touch of the yurt, as well as the integration of the grassland environment. Buhe said: "I think the combination of yurt and nature gives people a subtle feeling, such as the smell of wood, leather, felt, and the form of space. The house now has no such smell".

The Mongolian herdsmen's affection for freedom and nature stems from their long-term freedom of nomadic life, and is also reflected in their bold, homey, and unrestrained character. Alateng Aode said: "The nomadic lifestyle is a very free state of life. There is no fixed residence, so the herders will not worry about that residence . . . The Mongolians have their own freedom . . . It reflects a feeling among the people". Although the settlements have yurts and stable living areas, the Mongolians still choose to graze, and emotionally prefer the nomadic life on the grasslands. Ayongga said: "The yurt in the yard feels very different from the yurt in the pastoral area . . . Pastoral life is a very free state; on the

contrary, everything is settled after I settle down. I am more yearning for life in the pastoral area. I prefer a wide area".

*3.4. Residential Characteristics*

The category of residential characteristics is composed of three subcategories: "yurt space", "structure and material", and "physical environment".

The space attributes of the yurt are the adaptation of Mongolian nomadic life and living habits for thousands of years to the environment and including space scale, space layout, space quality, and order. The scale is appropriate, single and complete, facilitates communication, and has clear divisions. Zhaola (a 21-year-old Mongolian herder) said: "The round shape in our Mongolian [yurts] means unity and cohesion", as shown in Figure 6a–c.

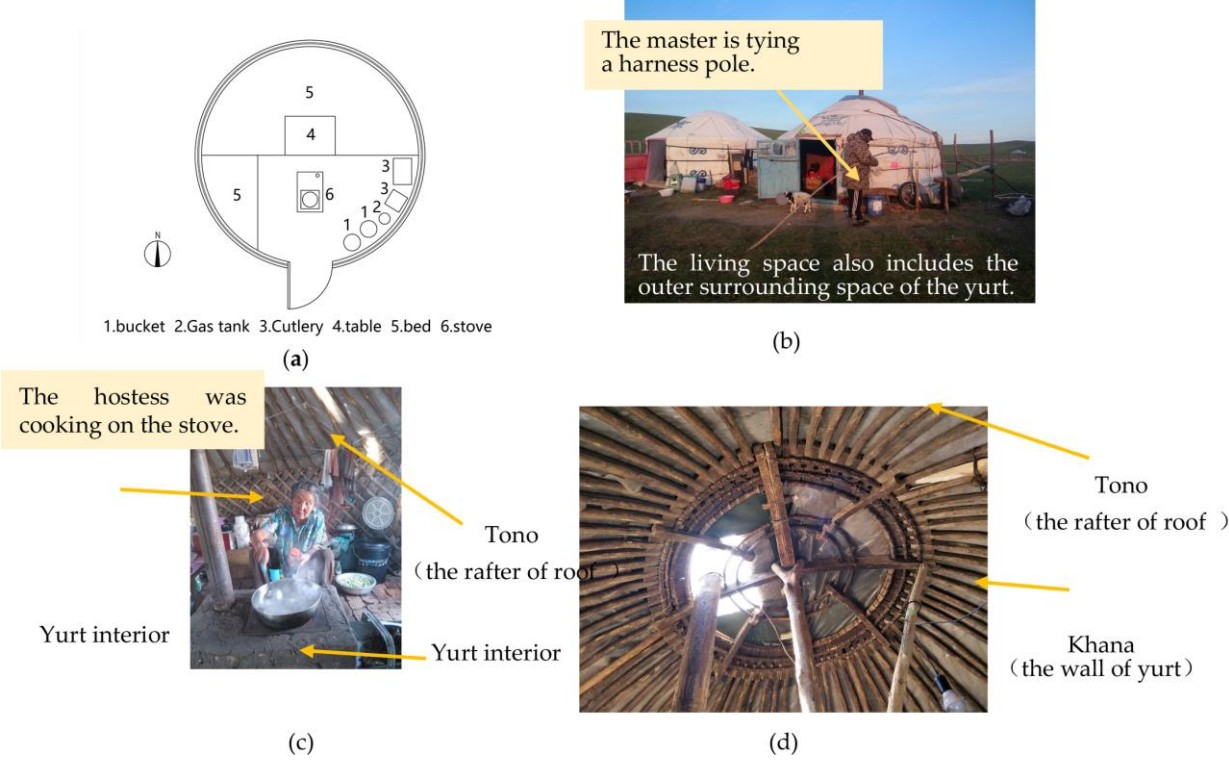

**Figure 6.** Residential characteristics: (**a**) yurt layout; (**b**) external layout of the yurt; (**c**) function; (**d**) structure and materials.

Structure and materials are important material factors in the construction of the yurt. In contrast to those living in other local residential houses, Mongolians are both users and builders. Traditional yurts are constructed with small-sized components to form a prefabricated wooden structure system, which is light in weight and easy to build. Most Mongolians and even the elderly over 60 can build a yurt by themselves in one hour, as shown in Figure 6c. Nowadays, iron yurts require 2~3 people to build over 1~2 h. The usual process for the foundation of a traditional yurt is to level the ground, spread sheep dung, and then cover it with felt. Iron yurts are built on bare ground or lawn, or laid with red bricks, cement floor tiles, and cast-in-place concrete. It can be seen that different materials and structures will affect the construction of the yurt.

The comfort of the yurt often needs to be adjusted through active behavior. Yurts have poor warmth retention. Traditional yurts were heated by burning stoves, and later burning kang (A kind of bedding that can be heated by fire, is a kind of house building facilities popular in northern China and Mongolian countryside for heating and resting sleep). A 74-year-old Mongolian herder recalled: "When I was young, I lived in a yurt,

which was extremely cold, so that I need to cover it with two layers of wool felt to sleep at night". Herders often adjust the enclosure interface of the yurt to achieve ventilation. In the scorching summer, they lift up the felt on the bottom edge of the yurt and open it to ventilate and cool down, as shown in Figure 6d. Moreover, the interior of the yurt is filled with the natural smell of pasture, cattle and sheep, which creates a special living atmosphere and reflects the mutual integration of the yurt and the external grassland environment.

The spatial order of the yurt is manifested as "sacred and secular boundaries, male and female divisions". As shown in Table 3, the yurt has a stove as the center, and the north of the stove is a divine space, where Buddhist niches, ancestors or portraits of Genghis Khan are placed; the south of the stove is a secular space, where daily production and living utensils are arranged; the west of the stove (right) is the men's area, and for men's daily activities mainly, so saddles, whips, telescopes, and other utensils will be placed in this part; the east of the stove (left) belongs to the women, and women's daily activities are mainly completed in the east area, so cooking utensils will be placed in this part.

### 3.5. Natural Environment

The category of natural environment is composed of two subcategories: "natural objects" and "relationship with natural environment".

The pasture environment is an important factor for nomadic life. It not only directly affects the condition of livestock, daily grazing, and care of livestock, but also indirectly affects the length and location of nomadic life. Uyoudai (a Mongolian herder) said: "Now the dry cattle on the pastures are not full of food. [We] even stop milking. The pastures were lush in the past, and their locations were different in spring, summer, and autumn. Herders can milk". For yurt dwellers, food, well water, and fuel (cow dung) and traditional yurt materials (willow wood, cowhide rope, wool felt) are all natural materials. The climatic environment, the seasons, and pasture conditions under its influence are the decisive elements for nomadic and yurt living. Daily weather conditions affect the behavior of yurt dwellings, travel, rest, window opening, door opening, and stove burning.

The daily production and life of herders is based on grazing of livestock, which is not only a source of food, but also a major source of income. Herders mentioned being busy in winter, picking lambs in spring, and cutting grass in autumn to prepare winter fodder. Zaragenbaier mentioned that he hopes to live with cattle, sheep, horses, and dogs (not just as pet dogs).

### 3.6. Consistency and Reliability Test of Results

According to Glaser and Strauss [64], theoretical saturation is the moment when it is impossible to obtain additional data to enable the analyst to further develop the characteristics of a certain category. Through supplementary interviews and related materials combing for theoretical verification, no new categories and characteristics appeared, so theoretical saturation was reached.

### 3.7. The Relationship between the Main Categories

In the structural relationship diagram of the five main categories, production-lifestyle is the core category, in which internal factors include cultural belief and perceptual experience; external factors include residential characteristics and physical environment. These categories constitute the main factors affecting the yurt residence behavior. The relationship between the main categories is shown in Figure 7.

#### 3.7.1. Lifestyle Is a Core Category

Production-lifestyle is the most essential, core factor, and the basic, decisive factor in other categories.

Mongolians are forced to choose a nomadic production method to survive in grassland, and have a simple lifestyle. The grassland policy will directly change nomad mode. Now there is underway a transition from nomadism to settlement, followed by a transitional

lifestyle of half-round and half-pastoral. Meanwhile, yurt living behavior and the core of "nomadism" has also changed.

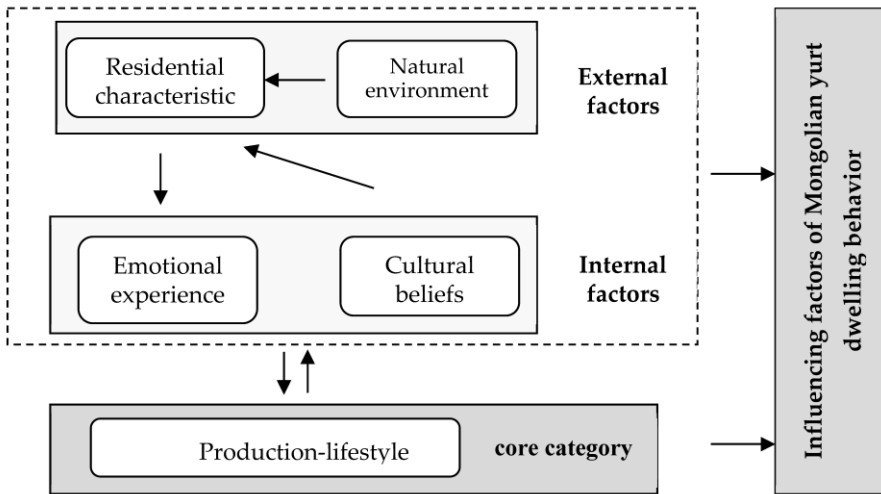

**Figure 7.** Diagram of the relationship between the main categories.

　　Production-lifestyle is the most essential and core factor, but also the basic and decisive factor in other areas. Alateng Aode proposed that "modern architects break the circular shape against the traditional production mode. The requirements of the nomadic production-lifestyle have led to the scattered settlement of Mongolians". Mongolian yurts are suitable for diaspora. In addition, as a fabricated felt tent building, the yurt is suitable for grassland nomads due to its convenient materials, easy construction, and easy relocation.

　　Nomadism affects the pasture environment, and a different nomad mode has an impact on the natural environment. Moreover, the specific living environment and production mode have nurtured cultural belief with national and regional characteristics. The special nomadic production-lifestyle for thousands of years has used yurts as the carrier to form the place meaning of the human and grassland environment, and cultivate the traditional emotional experience of Mongolians with collective perception and memory of grassland and livestock. Alateng Aode said: "Mongols have their own freedom of existence. The perennial nomadic life reflects a feeling among people".

### 3.7.2. Other Categories against Production Lifestyle

　　The ecological condition of the pasture, the growth of livestock, and the climate environment often have an impact on the nomadic production and lifestyle, which rely on the use of natural resources and are reflected in the time node, content, location, and duration. In addition, changes in yurt housing space functions, forms, structures, materials, and other residential characteristics will also cause changes in yurt building behavior, housework behavior, eating behavior, and other changes in residential life.

　　However, the production-lifestyle is related to material factors and spiritual factors. Specific national attributes and geographical environments form a unique culture and beliefs during historical accumulation, but in turn regulate the production-lifestyle of herders. Hada (a Mongolian college student) said: "Grandparents often teach us that children can't step over the stove, put their feet on the stove, and stand on the threshold . . . Taboo customs and rules, which are intangible, have been preserved".

　　This study found that compared with urban housing, herders prefer nomadic life in pastoral areas and prefer to live in round traditional wooden yurts. Buhe said: "The yurt combines with nature. It gives a subtle feeling". Zaragenmond said: "The yurt is not a necessity, but it gives me quiet feelings. I want to live occasionally".

### 3.7.3. Relationship between Other Main Categories

In terms of residential behavior in yurts, residential characteristics and the natural environment are external factors in objective sense, and emotional experience and cultural belief are derived from the interior factors of ideology and value concepts, both through behavioral activities.

The yurt is an ideal residential choice to adapt to the natural environment of the grassland. The residence characteristics of the yurt are the result of continuous optimization. It is manifested in functional layout, structural characteristics, material selection, and construction methods. In addition, the yurt residence has a spatial form, decoration, and connotation meaning of "building like the sky and respecting nature" under the historical accumulation. Compared with the cultural spirit, the natural environment has a greater impact on the characteristics of yurt dwellings. The reason is that yurt dwellings were originally designed to adapt to a specific natural environment.

The circular plane, dome, open interior space, natural enclosure materials, and openness of the yurt, these residential characteristics give people a special emotional experience. Zalaganbaier said that: "If you live in a yurt, you can feel the feeling of gathering, which is what we believe in".

## 4. Discussion

From the 1980s to the present, the change from nomadic residence to settlement occurred. For the Mongolian people, essentially a nomadic culture turned to a settled culture, and transformed herders' role from an open and free grassland adaptor to a fixed land adaptor. Facing the development of social informatization and the revitalization and development of China's rural areas, how to build a spatial and social environment that is more in line with the living and life of Mongolians, and to help them gain a sense of belonging, build cultural identity, and gain national self-confidence are important issues.

### 4.1. Lifestyle Is the Most Essential, Most Core Factor

From the perspective of Marx's philosophy, productivity determines production relationships. The production method of nomads determines the form of nomadic residence [65]. The Mongolian yurt was created by the nomads of the grassland during long-term migration, and was shaped after continuous exploration and gradual improvement. It carries the various appearances of the Mongolian material life, but also a comprehensive manifestation of the development level of material production [66]. Compared with the separation of the life circle and the production circle of the farming civilization, the grazing of the nomadic civilization is consistent with the place and environment of the dwelling. Therefore, the nomadic civilization production method has a more significant impact on residential life. The yurt was produced due to the emergence of nomadic activities, and died out due to the end of nomadic activities [15].

### 4.2. Inherent Influencing Factors—Cultural Belief and Emotional Experience

Environmental psychology believes that under the guidance of irritation or some purpose, body action produces empirical perception and intuition, thereby meeting the corresponding needs of space and environment. Consequently, emotional experience and cultural belief affect residence from the idea level and the perceptual level.

- Cultural belief

Residences premises to meet physiological and spiritual needs of life in the most direct and most realistic way. Especially, living behavior in traditional dwellings is the self-presentation of different races, nationalities, and regional groups. For specific nomadic and grassland environments, their fundamental determinants come from national collective awareness. Cultural spirit is the attitude towards life and the environment under the collective consciousness, which are manifested in residential behaviors including sacrificial activities, ritual behaviors, building behaviors, environmental protection behaviors, and

restrained grazing behaviors etc. An important part of culture—rituals—often acts as a "cultural map", guiding people's behaviors and making them conform to the requirements of social organization and structure in the process of dissemination [67]. Cultural belief is the core control factor for the continuation and development of yurt dwelling behavior, with a deep sense of living awareness, emotional value, and values.

Buildings and settlements are the visual expression of the priorities of life and the real world. As small as a house or house, as large as a village or a town, they all reflect the goals and life values shared by a specific society [2]. The form and internal layout of the Mongolian yurt, as well as having a special design according to temporal and spatial meaning, is the embodiment of the Mongolian group for the understanding of the universe. Even in the context of change, the Mongolian still adheres to the fear of nature, advocating freedom of ecological concepts and values, which have important enlightenment and reference significance for human survival and sustainable development.

- Emotional experience

The change of nomadic life to settlement has brought changes in social relationships and survival, causing a lack of Mongolian belonging and identity. However, these changes have strengthened the Mongolian people's emotional inclination towards yurt settlement and grassland environment, and aroused their collective emotional memory. Compared to noisy, intensive, concrete city space, Mongolian herders prefer freedom, relaxed, unobscured, quiet pastoral life. Psychology and sociology hold that almost any aspect of human cognition, behavior, and social organization is driven by emotion, and human behavior and consciousness are also affected by corresponding emotional feedback [68]. CASS et al. indicated that the attitudes and behaviors of individuals or groups are significantly affected by the emotions they assign to specific places [69]. The Mongolian people's yearning for living in a yurt is inseparable from emotional factors. They pay more attention to sensory experience, and pay more attention to taste and touch.

When building a residence for Mongolian herders, it is necessary to consider the basic conditions of space morphology, functional needs, and physical environment to meet residence needs. Moreover, it should meet the Mongolian people's emotional needs for freedom, broadening of horizons, and sense of belonging that are extremely valued.

### 4.3. Residence Characteristics of the Yurt

Amos Rapopor pointed out that the formation of a behavioral model is reflected in the behavioral mode; once the form is generated it in turn affects behavior and lifestyle [70], and the residence characteristics and behavior of Mongolian yurt dwellers show the relationship between two-way interaction.

The residence behavior of different nations is different in terms of customs and cultural traditions. Residential buildings, as a type of building closely related to people's daily life, can best reflect this difference [71]. The yurt is a flower that blooms in the soil of nomadic life. It is the most representative house of the Mongolian and northern nomads [15]. In the nomadic economy, due to the limitation of pasture and stocking capacity, the grazing households are scattered and far apart. As a result, the yurt has become the center of the shepherd's life and existence. The Mongolians cherish the yurt as their own life [3].

The distinctive residential characteristics of the yurt also have an important relationship with the cultural spirit. The spatial form of residence corresponding to the form of residential life is the result of a combination of many factors, among which social-cultural aspects are the primary factors [2]. The yurt shows the satisfaction and tendency of the Mongolian people to prefer the life of the grassland yurt from the aspects of adaptability of production and life, the integration of the grassland environment, the symbolism of cultural spirit, and the emotional experience. Ji Ya (a student from Mongolia) said: "The yurt can directly connect with the outside world, as if it is closer to nature. Compared with the modern square house, the round yurt will not have a sense of restraint".

## 5. Conclusions

The decisive influencing factors of Mongolian housing behavior include production-lifestyle, natural environment, residential characteristics, emotional experience and cultural belief. Production-lifestyle is the core category, residence characteristics and natural environments are external factors, while cultural belief and emotional experience are interior factors.

Production-lifestyle is the most basic, most essential factor affecting residence, interacting with others including natural environment, cultural belief, emotional experience, and residence. The pastoral policy is the fastest-acting, most distinct, most direct factor, changing the form and time of the nomadic life. Mongolian residence characteristics are the results of production-lifestyle, natural environment, cultural belief, and emotional experience. Emotional experience and cultural belief affect residence behavior from the thinking level and the perceptual level. Cultural beliefs have a deeper level and show the significance of collective consciousness in residential behavior.

Compared with urban homes, Mongolians have a high satisfaction with the grassland life and residence, which is manifested in the adaptability of the production and life of the yurt, the integration of the grassland environment, the symbolism of cultural spirit, and the emotional experience.

From the perspective of this research, the new yurt design can be further improved in terms of the diversity of yurt space, the extensibility of space, flexible use and infrastructure, and special attention should be paid to its cultural connotation and emotional experience. In addition, it is necessary to consider the construction, convenience, cost, and transportability of the yurt.

**Author Contributions:** Conceptualization, J.C.; methodology, J.C. and W.S.; software, J.C.; validation, L.B.; formal analysis, J.C.; investigation, J.C.; resources, L.B.; data curation, J.C.; writing—original draft preparation, J.C., W.S. and L.B.; writing—review and editing, H.G.; visualization, H.G. and W.S.; supervision, W.S. and L.B.; project administration, L.B.; funding acquisition, L.B. All authors have read and agreed to the published version of the manuscript.

**Funding:** This research was funded by Special scientific research project of Shaanxi Provincial Department of Education (Grant No. 22JK0314) and the National Natural Science Foundation of China (Grant No. 51768049).

**Data Availability Statement:** Not applicable.

**Conflicts of Interest:** The authors declare no conflict of interest.

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
