# Peer review of "A Grounded Theory Approach to the Influence Mechanism of Residential Behavior among Mongolian Yurt Dwellers in China"

_buildings, doi:10.3390/buildings13051268_

Round 1
Reviewer 1 Report
I have reviewed this interesting article which deals with the ways of inhabiting and interpreting the yurt.
The topic is necessary since there is an appreciation of this kind of nomadic dwelling which defies contemporary and/or modern concepts of inhabitation. Few peoples in the world have their dwellings as a national symbol and besides there is very little scientific information on their detailed features.
The authors could perhaps improve their methodology by further discussing the graphic approach to the problems. Also adding more plans and depictions of the individual yurt and also of the compounds of, the settlements I am curious about if the whole settlement follows a kind of Feng Shui, this could make the manuscript more readable.
The authors should be more careful when quoting references like Rapoport and not Rapopor, which appears as Amos in the bibliography while this is his first name and he should be quoted Rapoport A. and not Amos R.
Also prof. Funo Shuji. is not correctly reported in the references, being Funo its surname.
The article is very simple and emotive and it contributes firmly to anthropology and building typology’s knowledge. The grounded theory is not very much justified in my humble opinion and perhaps it is not necessary or redundant or it should be better explained.
However, the article in summary seems complete and thoroughly researched, I miss that they could collaborate with Mongolian native researchers and add more terminology in Mongolian Language.
The conclusions are consistent but perhaps could be slightly be expanded by comparing with other nomadic dwellings in the neighbouring countries.
Graphs and plots are not very detailed and they could be more reader-friendly. Also please add tables with the figures and geographical distribution of existing gherts.
Summary of evaluation: This article is interesting from the point of view of social sciences applied to buildings, it is also important as an area study for Mongolian rural areas. My suggestion is that the manuscript might published after some mild amendments.
Author Response
Thank you for your comments and suggestions on this paper. We did not expect that you have reviewed this paper so carefully. We have Revised the paper, please see the attachment for details: Response to Reviewer 1 Comments and Revised Manuscript.

Reviewer 2 Report
All notes and comments for the author were written on the original manuscript in the attached file

Author Response
Thank you for your comments and suggestions on this paper. We did not expect that you have reviewed this paper so carefully. We have Revised the paper, please see the attachment for details: Response to Reviewer 2 Comments and Revised Manuscript.

Round 2
Reviewer 1 Report
I thank the authors for effectively solving all my queries. This is now an excellent paper and deserves publication in Buildings as far as I know
Reviewer 2 Report
It appears that the majority of the comments we made during our previous review have been included in the revised manuscript. I think the manuscript has improved enough to merit publication in Buildings Journal.